# The Effects of a Meldonium Pre-Treatment on the Course of the Faecal-Induced Sepsis in Rats

**DOI:** 10.3390/ijms22189698

**Published:** 2021-09-08

**Authors:** Siniša Đurašević, Aleksandra Ružičić, Iva Lakić, Tomislav Tosti, Saša Đurović, Sofija Glumac, Slađan Pavlović, Slavica Borković-Mitić, Ilijana Grigorov, Sanja Stanković, Nebojša Jasnić, Jelena Đorđević, Zoran Todorović

**Affiliations:** 1Faculty of Biology, University of Belgrade, 11000 Belgrade, Serbia; a.ruzicic@bio.bg.ac.rs (A.R.); djiva@bio.bg.ac.rs (I.L.); jasnicn@bio.bg.ac.rs (N.J.); jelenadj@bio.bg.ac.rs (J.Đ.); 2Faculty of Chemistry, University of Belgrade, 11000 Belgrade, Serbia; tosti@chem.bg.ac.rs; 3Institute of General and Physical Chemistry, University of Belgrade, 11000 Belgrade, Serbia; sasatfns@uns.ac.rs; 4School of Medicine, University of Belgrade, 11000 Belgrade, Serbia; sofijaglumac09@gmail.com (S.G.); zoran.todorovic@med.bg.ac.rs (Z.T.); 5Institute for Biological Research “Siniša Stanković”—National Institute of Republic of Serbia, University of Belgrade, 11000 Belgrade, Serbia; sladjan@ibiss.bg.ac.rs (S.P.); borkos@ibiss.bg.ac.rs (S.B.-M.); iligri@ibiss.bg.ac.rs (I.G.); 6Centre for Medical Biochemistry, University Clinical Centre of Serbia, 11000 Belgrade, Serbia; sanjast2013@gmail.com; 7Faculty of Medical Sciences, University of Kragujevac, 34000 Kragujevac, Serbia; 8University Medical Centre “Bežanijska kosa”, University of Belgrade, 11000 Belgrade, Serbia

**Keywords:** sepsis, liver, kidney, heart, inflammation, oxidative stress, lipidomics, rats

## Abstract

Sepsis is a life-threatening condition caused by the dysregulated and overwhelming response to infection, accompanied by an exaggerated pro-inflammatory state and lipid metabolism disturbance leading to sequential organ failure. Meldonium is an anti-ischemic and anti-inflammatory agent which negatively interferes with lipid metabolism by shifting energy production from fatty acid oxidation to glycolysis, as a less oxygen-demanding pathway. Thus, we investigated the effects of a four-week meldonium pre-treatment on faecal-induced sepsis in Sprague-Dawley male rats. Surprisingly, under septic conditions, meldonium increased animal mortality rate compared with the meldonium non-treated group. However, analysis of the tissue oxidative status did not provide support for the detrimental effects of meldonium, nor did the analysis of the tissue inflammatory status showing anti-inflammatory, anti-apoptotic, and anti-necrotic effects of meldonium. After performing tissue lipidomic analysis, we concluded that the potential cause of the meldonium harmful effect is to be found in the overall decreased lipid metabolism. The present study underlines the importance of uninterrupted energy production in sepsis, closely drawing attention to the possible harmful effects of lipid-mobilization impairment caused by certain therapeutics. This could lead to the much-needed revision of the existing guidelines in the clinical treatment of sepsis while paving the way for discovering new therapeutic approaches.

## 1. Introduction

Sepsis is a life-threatening condition caused by the dysregulated and overwhelming response to infection, accompanied by the increased sequential organ failure assessment score in clinical settings [1,2]. In addition, the exaggerated pro-inflammatory state may cause multiple organ injuries in response to damage-associated molecular patterns (DAMPs) molecules derived from damaged host cells [3]. A cardiovascular collapse unresponsive to fluid resuscitation may be developed in the most severe cases of septic shock, urging vasopressors therapy [4]. According to Vincent et al. [5], for the 2005–2018 period, the ICU-, hospital-, and 28/30-day mortality of patients from North America and Europe was 37.3%, 39.0%, and 36.7%, respectively. In 2017, an estimated 48.9 million incident cases of sepsis and 11 million sepsis-related deaths were recorded worldwide, accounting for 19.7% of total global deaths [6].

Energy production appears to be very important for the prognostic assessment of sepsis [7]. Notably, it has been postulated that organ failure in sepsis is related to ischemia-induced tissue injury, but today it is well known that a normal amount of oxygen is delivered to the tissues during sepsis [8,9]. Decreased ability of tissues to use oxygen despite its normal availability gives rise to the theory of the impaired energy homeostasis in sepsis as a kind of metabolic ischemia [10]. Furthermore, it has been shown that tissue oxygen consumption is negatively correlated with the course of sepsis [11], and that a decrease in body core temperature may serve as an important marker of sepsis severity [12].

The main cause of the energy homeostasis impairment in sepsis is the mitochondrial dysfunction and damage [11], with several possible underlying mechanisms such as alteration of mitochondrial functions due to the increased radical oxygen species (ROS) production [13]. A shift from glucose to lipid metabolism has been demonstrated, followed by increased tissue lipids level, suggesting the role of lipids as the main energy source in sepsis [14]. However, it should be noted that increased lipolysis is often accompanied by the simultaneous disturbance of free fatty acids (FFAs) β-oxidation [15], thus causing lipotoxicity [16] due to the toxic lipid intermediates accumulation in tissues [17].

In our previous work, we showed that meldonium, an anti-ischemic drug clinically used to treat myocardial and cerebral ischemia [18], also expresses strong anti-inflammatory effects in the liver [19] and renal [20] ischemia/reperfusion. By decreasing the long-chain FFAs displacement from the cytosol into mitochondria and redirecting them to peroxisomes, meldonium inhibits L-carnitine biosynthesis and transport [21]. In peroxisomes, long-chain FFAs are then, in a carnitine-independent manner, metabolized to medium- and short-chain metabolites before further oxidation in mitochondria [22]. In this way, meldonium decreases the accumulation of toxic long-chain FFA intermediates in mitochondria, thus reducing the risk of mitochondrial injury [23]. In addition, meldonium upregulates the expression of the peroxisome proliferator-activated receptor γ coactivator 1α (PGC-1α) [24], a transcriptional coactivator of the genes encoding proteins that participate in mitochondrial biogenesis and function [25].

All the above considered, we hypothesized that pre-treatment with meldonium could be an effective therapeutic approach in preventing mitochondrial dysfunction and impaired energy homeostasis present in sepsis [13]. To address this issue, rats were pre-treated for four weeks with meldonium in 300 mg/kg b.m./day dosage and underwent sepsis by a single intraperitoneal injection of faeces (0.5 g faeces/1 mL saline/100 g b.m.). The PGC1-α expression, as well as carnitine, glycerol, triglycerides (TGAs), glucose, fructose, sucrose, lactic acid, and FFAs concentrations were measured in the liver, kidney, and heart of experimental animals, as a general marker of meldonium metabolic and lipidomics action.

Since adrenergic stimulation is necessary for efficient lipid metabolism, the adrenaline (ADR) and noradrenaline (NOR) concentrations were measured in serum and adrenal glands. The extent of tissue injury was assessed by tissue histology, and by measuring serum troponin T concentration and alanine aminotransferase (ALT), aspartate aminotransferase (AST), and alkaline phosphatase (ALP) activity levels. The degree of tissue apoptosis and necrosis were evaluated by measuring the liver, kidney, and heart Bax/Bcl-2 ratio as an apoptotic marker, and the tissue and serum level of high-mobility group box 1 protein (HMGB1) as a necrotic marker. The HMGB1 protein is considered a specific marker of sepsis due to its role in inflammatory progression [26,27]. Both clinical and non-clinical studies of sepsis have found that the expressions of HMGB1 are in a close association with the nuclear factor kappa-light-chain-enhancer of activated B cells (NF-kB) signalling pathways activation in the cells [28]. Consequently, sepsis-associated tissue inflammation was assessed by measuring levels of an activated form of NF-kB p65 (p-NF-κB p65) in the liver, kidney, and heart, and by measuring the liver level of haptoglobin (Hp), as one of the major acute-phase proteins with anti-inflammatory and antioxidative properties whose expression is regulated by p-NF-κB p65. The liver, kidney, and heart oxidative statuses were assessed by measuring tissue activity of copper-zinc superoxide dismutase (CuZnSOD), manganese superoxide dismutase MnSOD, catalase (CAT), glutathione peroxidase (GSH-Px), and the level of lipid peroxidation (thiobarbituric acid-reactive substances, LPO).

## 2. Results and Discussion

### 2.1. Survival Analysis

Monitoring of the survival rate and the rectal temperature of the animals was conducted every hour for nine hours after the induction of sepsis. (Figure 1). Contrary to our expectations, the mortality of animals in the M + S group was far higher (50%, or 8/16 animals) compared with the S group (12.5%, or 2/16 animals), as confirmed by Log-Rank significance *p* < 0.007, and Holm-Sidak overall significance *p* < 0.04. Rectal temperature recordings did not show a statistically significant difference between any of the experimental groups. We observed shallow and rapid breathing of animals just before their death, assuming cardiac arrest as the possible cause of death.

### 2.2. AST, ALT, ALP and Troponin T Serum Analysis

To examine the extent of organ injury, serum levels of AST, ALT, ALP, and Troponin T were analyzed. AST, ALT, and ALP are enzymes with a key role in the amino acid metabolism in a variety of organs, including liver, brain, kidneys, pancreas, bile duct, gallbladder, spleen, heart, and seminal vesicles [29]. Serum levels of AST, ALT, and ALP are assessed, e.g., as a part of a liver function test since liver injury is followed by the release of these enzymes into the bloodstream. Troponin T is a well-established marker of a heart attack as its serum concentration increases when the damage of a heart muscle is present. Present results showed that the serum concentration of AST, ALT, and ALP was increased in the S group (Table 1). This increase of AST and ALT, but not ALP, was efficiently prevented by the meldonium pre-treatment in the M + S group. Similarly, the increase in the serum level of Troponin T in the S group was reduced by 40% in the M + S group. Considering that elevation in ALT and ALP denotes a hepatocellular injury, whereas AST and Troponin T are markers of heart injury presence, it can be concluded that meldonium exerted protection against both organ injuries.

### 2.3. Histology Analysis

To further explore the extent of organ injury, we examined the liver, kidney, and heart histology (Figure 2, Table 2).

Liver histology of control animals revealed intact lobular architecture with the exception of the mild lymphocyte infiltration present in the portal spaces (Figure 2A, Table 1). However, the hepatic damage observed in both S and M + S groups was far greater in comparison with the control animals, with the changes much more prominent in the S group (Figure 2B–D, Table 2). The M + S group showed mild to moderate disorganization of the hepatic cords, moderate vacuolization, mild reactive changes such as spotty liver necrosis, moderate Kupffer cell hyperplasia, insignificant portal tract inflammation, and marked venous congestion (Figure 2D, Table 2). Spotty liver necrosis stands for the necrosis of the minute hepatic clusters, whereas the Kupffer cells activation accompanies the loss of liver metabolic function. In the same group, occasional scattered fatty changes were present in the hepatocytes adjacent to the central vein. An increased number of binucleated hepatocytes was also observed in samples from the M + S group, suggesting hyperplasia due to tissue regeneration. The main findings in the S group include moderate disorganization of the hepatic cords, portal inflammation, and necrosis involving larger groups of hepatocytes within the lobules (patchy necrosis) (Figure 2B,C, Table 2). Steatosis was more pronounced in the S group compared with the M + S group. The S group also exhibited diffuse parenchymal vacuolization with the irregular nuclear contours owing to the hydropic degeneration and the hepatocyte cytoplasm swelling.

Regarding kidney histology, the control animals demonstrated rare lesions in comparison with both the S and M + S groups (Figure 2A1–D1; Table 1). Two hallmarks of acute tubular injury, i.e., loss of the epithelial brush border (the microvillus brush border normally present on the luminal surface of epithelial cells of the proximal tubule), and flattening of the epithelial cells were present to various extent in both the S and M + S groups, although more often in the M + S group (Figure 2B1–D1; Table 1). Evaluation of distribution and intensity of more severe kidney lesions, such as tubular necrosis, tubular lumen obstruction, and bleb formation showed worsening as a result of meldonium pre-treatment, with twice as many changes in the M + S group compared with the S group. Tubular necrosis, which plays a major role in the pathogenesis of acute tubular damage, is often accompanied by the rupture of the basement membrane (tubulorrhexis), and occlusion of tubular lumens by casts (tubular lumen obstruction). Cells subjected to noxious stimuli develop membrane deformations, termed plasma membrane bleb formation, which may cause organ-specific injury. For example, blebs proximal to the renal tubule cells may break off into the tubular lumen, thereby increasing the chances of tubular cast formation and tubular lumen obstruction. Besides, glomerular tubularization (i.e., the result of retrograde growth of proximal tubular cells along the luminal surface of Bowman’s capsule) was also more frequent in the M + S group than in the S group, indicating more severe kidney injury.

The heart analysis showed no significant alterations in control animals, except for the presence of discrete foci of myocytolysis localized predominantly subendocardially (Figure 2A2, Table 1). Myocytolysis, also termed ‘vacuolar degeneration’ or ‘colliquative myocytolysis’, is characterized by the intracellular vacuolization and a loss of cross striations. The seemingly empty spaces are possibly filled with water, glycogen, and/or lipids, causing cell death and tissue scarring. Evidence of the heart remodelling, such as hypertrophy and dilatation, were also rarely present in the control animals. Conversely, ca. 50% of the S group samples showed presence of the interstitial lymphocytic cell infiltrates localized at the edges of the myocytes (Figure 2B2, Table 2). Focal loss of myocytes, hypertrophy, and dilatation were much more prominent in the S group in comparison with the control group. However, in the M + S group, vacuolar degeneration and interstitial mononuclear infiltration were even more widespread along with a greater loss of myocytes (Figure 2C2–D2, Table 2). Presence of the myofibrillar contraction band necrosis was also observed in the M + S group. In addition, the M + S group displayed widenings of the myocardial interstitium and abundant oedema, caused by the fluid accumulation. Observed cytoplasmic vacuolization may correlate with the presence of the lymphocytic infiltrates, although it could represent a nonspecific change since it has been previously associated with ageing, ischaemia, and/or poor ventricular function of the unknown aetiology. The aforementioned contraction band necrosis develops as calcium ions enter the cardiomyocytes through the damaged cell membrane, causing hypercontraction of the myofibrils which then form bright eosinophilic cross-bands.

As we summarize the histological findings, it can be observed that meldonium exerted a protective effect in the liver, while actually worsening the histological score in the kidney and heart. On that account, it could be assumed that sepsis induces acute kidney injury by reducing the renal blood flow. However, recent studies showed that in sepsis, normal or even higher cardiac output is present, followed by the systemic vasodilatation which further maintains the physiological rate of renal blood flow [30]. The same is true for meldonium, as it is known to increase the blood flow and consequently the oxygen flow throughout the body. Therefore, our results that indicated acute kidney injury in the M + S group were completely unexpected. The worsening of the heart histological score in the M + S group was also surprising. It should be noted that depression of the cardiac function in sepsis is quite common, with the pathogenesis of septic cardiomyopathy being complex and most likely multifactorial, including metabolic alterations such as decreased glucose uptake and increased lactate production [31,32]. Our previous studies show an increase of the glucose uptake and decreased lactate production as a result of the meldonium pre-treatment in both liver [19] and kidney [20] models of ischemia/reperfusion injury. Accordingly, we expected meldonium to exert protective effects in the present study as well.

### 2.4. Assessment of the Inflammatory Status

To explain the histological findings as well as the higher mortality rate of the M + S group, we performed an analysis of the tissue inflammatory status. As a dysregulated inflammatory response, sepsis is manifested by the excessive release of inflammatory mediators, such as HMGB1, from the innate immune cells [26,33]. Actively released HMGB1 serves as an alarming signal to recruit, alert, and activate innate immune cells, thereby sustaining a potentially injurious inflammatory response [33]. We report that the S group exhibited a significantly increased level of HMGB1 in the serum and in the liver, kidney, and heart whole homogenates (1.3-, 1.2-, 1.4-, and 1.7-fold, respectively) (Figure 3A–D). Conversely, in the M + S group, this sepsis-induced increase was reduced by 23% in serum, 21% in liver, 34% in kidney, and by 18% in heart (Figure 3A–D).

Since the underlying mechanism of the proinflammatory action of HMGB1 includes activation of NF-κB signalling pathways [28], we examined the changes in the NF-κB p65 activation, using Western immunoblot analysis of the liver, kidney, and heart whole homogenates with an antibody against the phosphorylated form of this protein. As shown in Figure 3B,C, the S group demonstrated an increase of the serum and tissue HMGB1 level followed by an increase of p-NF-κB p65 level in the liver (1.7-fold), kidneys (1.6-fold), and heart (1.4-fold). Simultaneously, meldonium-induced serum and tissue reduction of HMGB1 levels in the M + S group was followed by a decrease in p-NF-κB p65 level in the liver (25%), kidneys (34%), and heart (19%).

The NF-κB p65 mediates transcription of a large number of protein targets involved in septic pathophysiology, including Hp and other damage-associated molecular patterns (DAMPs) proteins [34]. As an acute-phase protein, Hp is predominantly synthesized in the liver in response to different pathological stimuli and inflammatory mediators [35,36], such as those present in sepsis [37,38]. Hp is released from damaged tissue, which is why its increased level implies extensive cell death [39]. Our study revealed that the hepatic activation of NF-κB p65 in the S group also leads to the concurrent 4.7-fold increase in the hepatic Hp level. On the contrary, hepatic p-NF-κB p65 decrease in the M + S group led to a decrease of the Hp level by 55% (Figure 3B).

Furthermore, we examined major markers of apoptosis, as it is tightly linked to inflammation and sepsis-related mortality. Inflammation-activated NF-κB p65 signalling is involved in the regulation of cell apoptosis in different organs [40]. Mechanisms underlying apoptosis include a delicate interplay of pro-apoptotic and anti-apoptotic members of the Bcl2 protein family, such as Bax and Bcl-2 proteins, with the latter being able to block both apoptosis and necrosis [41]. Consequently Bax/Bcl-2 ratio is considered an apoptotic marker as its raised values support the presence of proapoptotic events, whereas lowered levels suggest anti-apoptotic events. In our study, the Bax/Bcl-2 ratio was increased 3-fold in the liver, 2.5-fold in the kidney, and 1.4-fold in the heart of animals from the S group, indicating increased tissue apoptosis in sepsis (Figure 3B–D). In the M + S group, its ratio was decreased by 39% in the liver and kidneys, and by 32% in the heart, suggesting that, altogether, meldonium exerted anti-apoptotic effects. We stress that the changes in the cardiac Bax/Bcl-2 ratio could explain the observed changes in serum Troponin T levels presented in Table 1. Namely, cardiac Troponin T is released into serum following the myocardial injury [42], caused either by necrosis or apoptosis [43]. Therefore, the increased serum Troponin T level in animals from the S group and the decreased serum Troponin T level in animals from the M + S group (Table 1) both correspond with the increased cardiac Bax/Bcl-2 ratio in animals from the S group, and the decreased cardiac Bax/Bcl-2 ratio in animals from the M + S group (Figure 3D).

Present results suggest that meldonium is able to ameliorate inflammatory reactions following sepsis induction, with the activation of HMGB1/NF-κB signalling as a proposed underlying mechanism. A similar observation was made by Xing et al. as they investigated beneficial roles of oleuropein in sepsis-induced myocardial injury [44]. Therefore, the effects of meldonium on sepsis-related inflammatory changes cannot explain the higher mortality rate observed in the M + S group.

### 2.5. Assessment of the Oxidative Status

Increased production of reactive oxygen species (ROS) is a well-recognized prognostic marker of sepsis outcome [13]. ROS production is tightly linked to mitochondrial dysfunction and damage [11,45]. So far, it was shown that several antioxidative enzymes could be upregulated by p-NF-κB. However, its role in regulation of the antioxidative defence is still extensively investigated. For instance, MnSOD is one of the well-known NF-κB targets [46,47]; however, its proposed targets also include CuZnSOD [48], catalase [49], and GSH-Px [50].

As seen in Figure 4, the aforementioned increase in the hepatic p-NF-κB level of the S group (Figure 3B) was followed by an increase in the hepatic GSH-Px activity. Additionally, this group exhibited an increase in the level of lipid peroxidation, which may contribute to the observed GSH-Px activity increase, as it is known that besides its role in H_2_O_2_ conversion GSH-Px can also reduce lipid peroxides [51]. In hearts of animals from the S group, sepsis increased CuZnSOD activity, which can also be explained by the cardiac p-NF-κB increase (Figure 3D). However, in both the liver and heart, no differences were observed in GSH-Px and/or CuZnSOD activities when comparing S and M + S groups, despite a strong decrease in p-NF-κB levels in both organs under the meldonium pre-treatment (Figure 3B,D). Moreover, the cardiac lipid peroxidation in the M + S group was lower by almost 30% in comparison with the S group. In kidneys of animals from the S group, the changes in antioxidant enzymes activities were opposite to the findings observed in both the liver and heart. Namely, regardless of the sepsis-induced p-NF-κB increase (Figure 3C), the renal activities of CuZnSOD, CAT, and GSH-Px decreased, whereas the MnSOD activity increased. However, as in the case of the liver and heart, there were no differences in enzyme activities between the S and M + S groups.

The degree of oxidative stress assessed by the changes in the activity of some of the antioxidant enzymes must be interpreted with great caution, as these changes are highly dependent on the context in which they take place. For instance, the increase of a given enzyme activity could be interpreted as an increase in the antioxidant protection due to the more efficient ROS removal. However, the same change may also be interpreted as a mechanism compensating for the increased ROS production, therefore implying oxidative stress increase. Based on the present results, it can be concluded that oxidative stress was increased only in the liver of animals from both S and M + S groups, primarily due to the increase in lipid peroxidation as a direct indicator of oxidative stress. In the heart, there was also no evidence of such an increase, as the level of lipid peroxidation remained unchanged in the S group, and even decreased in the M + S group. In kidneys, changes in the activity of the antioxidant enzymes suggest increased antioxidant protection, and thus reduced oxidative stress. As previously reported, the increase in MnSOD activity can be perceived as a more effective way in the removal of the respiratory chain-derived superoxide anion radicals, hence reducing the propagation of oxidative stress from mitochondria to the cytoplasm, and reducing CuZnSOD activity [52]. Since superoxide anion dismutation results in the production of hydrogen peroxide, a reduction in CuZnSOD activity would lower H_2_O_2_ production, therefore decreasing the activity of both CAT and GSH-Px as well. Taken together, the present results did not provide evidence of antioxidant status deterioration, thus providing no explanation for the increased mortality in the M + S group (Figure 1).

### 2.6. Lipidomic Analysis

Given that sepsis is characterized by both systemic and organ-specific metabolic changes [53], we performed an extensive lipidomic analysis. As seen in Table 3, a similar pattern of increase was observed regarding carnitine, glycerol, lactate, and total concentration of examined FFAs in the liver, kidneys, and heart. Sepsis increased carnitine, glycerol, lactate, and total FFAs concentration, whereas all these parameters were strongly reduced in the meldonium pre-treated group (Table 3). In addition, glucose, fructose, and sucrose concentrations were increased in the liver and kidneys of the S group, and decreased in the M + S group. This mobilization of all available energy resources strengthens the hypothesis of “septic auto-cannibalism”, further promoting sepsis as a metabolic failure under the increased energy demand [54,55].

The observed decrease of tissue carnitine, glycerol, lactate, and total FFAs concentration in the M + S group is a result of the well-recognized meldonium effects. Namely, meldonium lowers L-carnitine concentration both by preventing its formation through the inhibition of γ-butyrobetaine hydroxylase activity and by preventing its renal reabsorption [56,57]. This explains a drop of TGAs hydrolysis products, such as glycerol and total FFAs concentration (Table 3). In addition, meldonium shifts energy production to glycolysis, which explains the decrease in liver and kidney glucose concentrations. Intriguingly, this meldonium-induced metabolic shift was followed by a decrease in lactate concentration. This implicates ATP production through the substrate-level phosphorylation, as previously reported in both liver [19] and kidney [20] models of ischemia/reperfusion injury. The present results suggested that meldonium decreases lipolysis and fatty acids β oxidation, and stimulates aerobic oxidation of glucose as an oxygen-sparing mechanism for ATP production under septic conditions.

Nevertheless, the examined organs did differ, to a certain extent, in their lipidomic profiles. For instance, the increase in the lipolytic demand of the S group was followed by the increased TAG concentration in both the liver and heart, but not in the kidneys. Similarly, the decreased lipolytic demand in the M + S group was followed by the decreased TAG concentration in the liver and heart, but not in the kidneys (Table 3). In this regard, it should be recalled that in the S group, total FFAs concentration was far higher in the kidneys when compared with that observed in the liver and heart. That is to say, renal FFAs concentration was already high enough to efficiently fulfil its energy demands, thereby surpassing the need for additional FFAs accumulation.

Even though total FFAs concentration was affected in all three examined organs, i.e., increased in rats from the S group, and the decreased in rats from the M + S group, organ-specific changes in concentrations of particular FFA were evident (Table 3). For example, the changes in tissue concentration of palmitoleic acid (16:1) were far more conspicuous in the kidney and heart compared to the liver (Table 3). While its tissue concentration in the M + S group was decreased in all three tissues, the rate of this decrease varied greatly among the organs, being 2-fold and 1.5-fold lesser in the liver of rats from the M + S group in respect to the rats from the S and C groups, but 123-fold and 615-fold lesser in the kidneys, and 30-fold and 142-fold lesser in hearts of animals from the M + S group in respect to animals from S and C groups. Palmitoleic acid (16:1) is recognized for its role in glucose homeostasis and insulin sensitivity improvement [58], the reason being its ability to increase skeletal muscle insulin-stimulated glucose uptake [59] and reduce hepatic steatosis, inflammation, and insulin resistance [60]. The mechanisms underlying palmitoleic acid action include inhibition of lipid synthesis through downregulation of fatty acid synthase and stearoyl-CoA desaturase 1 [61], promotion of white adipose lipolysis, and activation of nuclear receptor peroxisome proliferator-activated receptor α (PPARα) [62]. PPARα exerts effects such as promotion of FFAs β-oxidation and the reduction of overall ectopic triacylglycerol depots [63]. Therefore, a huge drop in kidney and heart palmitoleic acid concentration may have been sufficient to challenge energy production in the M + S group, thus leading to the increased mortality (Figure 1) and worsening of histological scores (Figure 2).

Odd-chain fatty acids (OC-FAAs), e.g., pentadecylic (C15:0) and margaric acid (C17:0) are of great importance for energy homeostasis [64]. Our results revealed that the decrease in tissue concentration of C15:0 was especially high in the kidney of animals from the M + S group (25-fold and 653-fold in respect to rats from S and C group), while the decrease in tissue C17:0 was especially high in the hearts of animals from the M + S group (60-fold and 145-fold in respect to rats from S and C group) (Table 3). Based on published data, the circulatory level of C15:0 and C17:0 negatively correlates to the incidence of the metabolic disease [65] and coronary heart disease risk [66]. Consequently, we assume that a strong decrease in kidney and heart C15:0 and C17:0 concentration observed in the M + S group (Table 3) may have contributed to the mortality rate increase (Figure 1) and worsening of the histological scores (Figure 2).

Sterol regulatory element-binding proteins (SREBP) are major transcription factors that regulate the expression of genes involved in FFAs and cholesterol biosynthesis [67]. It is known that oleic/elaidic (C18:1, cis and trans) acids may alter hepatic lipid metabolism, although in an opposite way. Elaidic acid upregulates hepatic de novo FFAs and cholesterol synthesis through SREBP expression increase, whereas oleic acid inhibits this action. Elaidic acid was also found to increase several lipogenic genes involved in the FFAs and sterol synthesis [68]. We observed a strong decrease in oleic/elaidic tissue concentration, especially in the kidneys (60-fold and 145-fold in respect to rats from S and C groups) and hearts (60-fold and 145-fold in respect to rats from S and C groups) of rats from the M + S group (Table 3). The present method for determining oleic/elaidic concentration allowed only for their simultaneous coelution, but nonetheless we can still assume that a significant decrease in their cumulative concentration in the kidneys and heart of the meldonium pre-treated animals interfered with energy production in such a way so as to increase animal death (Figure 1) and contribute to the reported histological findings (Figure 2).

Notably, the majority of the observed changes were linked to the kidneys and heart, thereby correlating with the energy expenditure of these organs. The heart secures approximately 60%–100% of its energy requirements via FFAs oxidation, while glucose and lactate account for the rest (Preau et al., 2021). Similarly, the kidney has one of the highest metabolic rates (Wang et al., 2010), as it consumes ca. 7% of the body’s daily ATP consumption [69].

### 2.7. Sympathoadrenal Activation

One of the leading factors participating in sepsis-induced organ injury includes sympathoadrenal overstimulation [70]. Our results showed that sepsis induced a decrease in adrenaline concentration in both adrenal glands and serum (40% and 25%, respectively, Table 3). The S group also exhibited a 15% increase in the serum noradrenaline concentration compared with the control group, whereas the adrenal glands’ noradrenaline concentration remained unchanged.

A decrease of adrenaline concentration under septic conditions was an unexpected result. Numerous studies report a strong activation of the septic-induced sympathetic drive in different organs, including the kidney, skeletal muscle, and heart. This is especially important in the case of septic cardiomyopathy [71], in which β-adrenergic blockade therapy showed promising therapeutic potential [72]. In addition, the sympathetic nervous system exerts a main role in lipid metabolism. Catecholamines, mainly adrenaline, activate signalling pathways responsible for the phosphorylation of key proteins in the process of white adipose tissue lipolysis [73]. Furthermore, it should be emphasized that during sepsis, an initial hypermetabolic, followed by a subsequent hypometabolic phase, can be observed [7]. The hypometabolic phase possibly has an adaptive role, during which cells reach a certain degree of metabolic hibernation, the reason being the protection against adenosine triphosphate depletion [4]. Therefore, a decrease in adrenaline concentration in serum and adrenals of the S group could be contributing to this adaptive response. Namely, a simultaneous rise in serum noradrenaline concentration may represent a compensatory sympathetic reaction opposing the serum adrenaline decrease, as previously reported in septic patients [74]. In the M + S group, serum adrenaline concentration was reduced by an additional 50% compared with the S group and remained unchanged in adrenal glands. Conversely, noradrenaline concentration was reduced in both serum and adrenal glands compared with the S group. It can be assumed that a strong reduction in the serum catecholamine levels was sufficient to disturb the lipid metabolism and increase mortality rate, as observed in the M + S group (Figure 1).

### 2.8. PGC1-α Protein Level Analysis

PGC1-α coordinates the upregulation of mitochondrial biogenesis and plays an important role in the metabolic reprogramming in response to the dietary availability by acting as an upstream regulator of lipid and glucose oxidative metabolism in a variety of tissues [75]. It has been suggested that meldonium could be involved in the PGC1-α upregulation by acting as an inhibitor of carnitine-palmitoyltransferase-1 (CPT-1) and thus leading to the accumulation of acyl-CoA and fatty acids in the cytosol [21]. Recently, Di Cristo et al. demonstrated that meldonium indeed increased PGC1-α expression on a *Drosophila* model of Huntington’s disease [24].

Our results showed that neither meldonium nor treatments significantly altered PGC1-α protein level in any of the examined tissues (Figure 5), in contrast to studies on different models of sepsis showing a reduction of PGC1-α content in the heart, kidney, and liver [76,77,78,79]. Since the proposed mechanism of PGC1-α downregulation under the septic conditions includes NF-kB activation [80,81], which we did observe, the supposed inconsistency between our results and those previously published could be explained by the difference in the experimental designs. Specifically, the experimental design mentioned in the literature implied tissue collection 12–72 h following the sepsis induction [76,77,78,79]. In contrast, we collected organs as early as within 8–9 h due to the high mortality rate following FIP injection. As expected, this time window induced strong NF-kB activation, but might not be sufficient to spot changes in the expression level of metabolic key players such as PGC1-α.

All the results considered, the question is whether the shown effects of meldonium are sepsis-independent, that is, whether meldonium causes the same effects in sham animals. We did not have a sham + meldonium animal group in this experiment, but we did include it in our previous experiments with the liver [19] and renal [20] ischemia/reperfusion. The results of both experiments clearly demonstrated that meldonium has no harmful effects in sham animals. In this regard, we conclude that the increased animal mortality observed in the M + S group was the result of the septic conditions, whereas in other invasive but non-septic experimental models meldonium still successfully exerts its protective effects. This is a very important conclusion, as it implies the importance of considering metabolic status of septic patients, especially of those treated with therapeutics primarily targeting the FFAs metabolism [82].

## 3. Materials and Methods

### 3.1. Animals and Treatments

All animal procedures were performed in compliance with the ARRIVE guidelines and Directive 2010/63/EU. Following the National legislation, all animal procedures were approved by the Veterinary Directorate of the Ministry of Agriculture, Forestry and Water Management, License number 323-07-05650/2021-05/1.

Sprague-Dawley strain (Rattus norvegicus) male rats weighing 329.9 ± 7.98 g were used for the experiment. The animals were acclimated to 22 ± 1 °C and maintained under a 12 h light/dark regime. The rats were randomly divided into three groups and housed two per cage for four weeks with *ad libitum* access to a standard diet (Veterinary Institute, Subotica, Serbia) and tap water (with or without meldonium).

Rat groups were as follows (Table 4): control sham group of animals that drank tap water for four weeks and then received a saline injection (C group); a septic group of animals that drank tap water without meldonium for four weeks, and then received a faecal intraperitoneal injection (S group); and the meldonium-septic group of animals that drank tap water with meldonium for four weeks, and then received a faecal intraperitoneal injection (M + S group). The experiment initially started with the eight animals in the control group and with 16 animals in S and M + S group to ensure that at least 8 survivors remained in these two groups in the event of the death of animals. At the end of the experiment, 8 surviving animals were randomly selected from the S and M + S groups and, together with 8 control animals, were decapitated for serum and tissue collection. The animals’ weight per group was 339.15 ± 20.51 g, 303.5 ± 15.01 g, and 348.44 ± 16.15 g for the C, S, and M + S group, respectively.

Meldonium (3-(2,2,2- trimetilhidrazinijum)) propionate, THP, and MET-88, manufacturer Shenzhen Calson Bio-Tech Co., Ltd., Shenzhen, China, were dissolved in tap water in concentrations ranging from 2–3 mg/mL. Depending on the water intake, meldonium concentrations in the water were adjusted weekly to achieve its consumption of around 300 mg/kg b.m./day. Based on the four-week measurement, the meldonium consumption in the M + S group was 296.97 ± 2.96 mg/kg b.m./day.

The day before the initiation of sepsis, faeces were collected from cages of control animals and put in 50 mL flasks (one flask per cage). Faeces were measured, and saline was added in a 1:1 ratio (1 g of faeces + 1 mL saline, final concentration 1g/mL). Flasks were left overnight on a rotator at 4 °C. The next morning, another volume of saline was added to the final concentration of 0.5 g/mL. Each flask was vortexed separately for 5 min until full homogenization. The content was filtered twice through quadruple gauze and stored at +4 °C until further use.

The faecal solution was administered by intraperitoneal injection to animals from the S and M + S groups at a dose of 0.5 g of faeces/mL of saline/100 g b.m. Animals from control groups received a sham injection with saline only (1 mL of saline/100 g b.m.). After the sepsis initiation, animals were monitored each hour for the rectal temperature.

### 3.2. Serum and Tissue Collection

Rats were euthanized by decapitation at any given time point between 8–9 h after the initiation of sepsis [83], and blood and tissue samples were collected immediately after euthanasia.

Blood was collected and incubated at room temperature for 45 min to allow clot formation. Then, the clot was removed by centrifugation at 2000× *g* for 10 min at 4 °C. The resulting supernatant was immediately transferred into a clean polypropylene tube using a Pasteur pipette [84].

The animals’ livers, kidneys, and hearts were isolated and dissected within 3 min, washed with the ice-cold 155 mmol NaCl, and placed in the same solution. One part of each tissue was placed in formaldehyde for further histological analysis. The serum and the rest of the tissue samples were stored at −80 °C for further analysis.

### 3.3. Biochemical Analysis

#### 3.3.1. Serum Analysis

Activities of ALT, AST, and AP in serum were measured by Roche Cobas C501 automated analyser (Roche Diagnostics, Mannheim, Germany), using ALTL, ASTL, and ALP2L reagent cassette.

Cardiac Troponin T level was measured with a highly sensitive assay based on electrochemiluminescence technology, using the Roche Cobas e601 automated analyser (Roche Diagnostics, Mannheim, Germany).

#### 3.3.2. Tissue Analysis of Lactic Acid, L Carnitine, Sugars, and Sugar Alcohols

Tissue samples (50 mg) were mixed with 2 mL of ultra-pure water and pulverized with a tissue grinder. After extraction in an ultrasound bath termostated at 0 °C, samples were centrifuged at 12,000 rpm. The supernatant was transferred in a 10 mL normal flask and diluted with ultra-pure water to the mark. The sample solutions were kept at −80 °C until analysis.

Tissue lactic acid determination: the lactic acid standard was purchased from Sigma Aldrich. Ion chromatography was used to assay the appearance and quantification of lactic acid. For that purpose, a Dionex ICS-3000 chromatographic set-up consisting of a single pump, conductivity detector (ASRS ULTRA II (4 mm) (P/N 061561), recycle mode), eluent generator (potassium hydroxide (KOH) (P/N 058900)) with Chromeleon^®^ Chromatography Workstation with Chromeleon 6.7.2 Chromatography Management Software was employed. The separation was performed on IonPac AS15 Analytical, 4 × 250 mm (P/N 053940) and IonPac AG15 Guard, 4 × 50 mm (P/N 053942) column. Mobile phase flow rate was set to 1000 mL/min, while the concentration of potassium hydroxide was changeable to achieve the following gradients: 0–15 min. 10 mM KOH; 15–25 min. 10–45 mM KOH; 2526 min. 45 mM KOH; 26–31 min. 45–10 mM KOH; and 31–36 min. 10 mM KOH. The column temperature was termostated at 30 °C, conductivity cell temperature was 35 °C, suppressor current was 134mA, and the backpressure was ~2200 psi.

Tissue carnitine determination: L-Carnitine standard was obtained from Sigma Aldrich. 10 mg of L-carnitine standard were weighted in a 10-mL normal flask and diluted with ultra-pure water to mark. Ascending thin-layer chromatography on RP-18 silica (Art. 5559, Merck, Germany) as adsorbents was performed. The chromatograms were developed using an acetonitrile-water binary mixture with a 1:1 volume ratio. The classical chromatographic chamber (Camag, Switzerland) was filled with 3 mL of mobile phase made by components of an analytical grade of purity. The chamber was saturated with solvent for 30 min. All experiments were performed on ambient temperature. The plates were applied with CAMAG Linomat 5 with 2 µL aliquots of previously prepared and defrosted aqueous solutions. The plates were scanned by CAMAG TLC Scanner 3 at 260 nm, and the obtained chromatograms were analyzed by winCATS software version 1.4.2.8121.

Tissue sugars and sugar alcohols determination: The standards of glucose, fructose, and sucrose were purchased from Tokyo Chemical Industry, TCI, (Europe, Belgium), while glycerol standard was obtained from Sigma-Aldrich (Steinheim, Germany). Sodium hydroxide and sodium acetate trihydrate were obtained from Merck (Darmstadt, Germany). All aqueous solutions were prepared using Ultrapure TKA deionized water. A standard solution of glucose was prepared in ultrapure water at 100 ng/mL concentration. Calibration standards were prepared from stock solution by dilution with ultrapure water. The quality control mixture used for monitoring instrument performance was prepared by diluting standard to concentrations of 20 ng/mL. Chromatographic separations were performed using DIONEX ICS 3000 DP liquid chromatography system (Dionex, Sunnyvale, CA, USA) equipped with a quaternary gradient pump (Dionex, Sunnyvale, CA, USA), a Carbo Pac^®^PA100 pellicular anion-exchange column (4 × 250 mm) (Dionex, Sunnyvale, CA, USA) at 30 °C. The mobile phase consisted of the following linear gradients (flow rate, 0.7 mL/min): 0–5 min. 15% A, 85% C; 5.0–5.1 min. 15% A, 2% B, 83% C; 5.1–12 min. 15% A 2% B, 83% C; 12–12.1 min. 15% A, 4% B, 81% C; 12.1–20 min. 15% A, 4% B, 81% C; 20–20.1 min. 20% A, 20% B 60% C; 20.1–30 min. 20% A, 20% B, 60% C; where A was 600mM sodium hydroxide, B was 600mM sodium acetate, and C was ultrapure water. Previously, the analysis system was preconditioned at 15% A and 85% C for 15 min. Each sample (25 µL) was injected with an ICS AS-DV 50 autosampler (Dionex, Sunnyvale, CA, USA). The electrochemical detector consisted of gold as working and Ag/AgCl as a reference electrode.

#### 3.3.3. Lipidomics Tissue Preparation

Tissue samples (150 mg) were mixed with 2 mL of chloroform/methanol (2/1, *v*/*v*) and pulverized with a tissue grinder. After extraction in an ultrasound bath termostated at 0 °C, 1 mL of 0.9% solution of NaCl was added. The samples were centrifuged at 12,000 rpm. The supernatant was transferred to a test tube and evaporated in a stream of nitrogen. The solid residues were dissolved in 3 mL of hexane, and 1 mL was kept for further analysis, whereas 2 mL were hydrolyzed with 1 mL of 2 M KOH solution in methanol. The mixture was put in an ultrasonic bath for 2 min at 70 °C. The excess of the potassium hydroxide was neutralized with 2 M HCl solution in methanol. To extract fatty acids, 3 mL of hexane were added to the mixture. The organic layer (hexane) was collected and evaporated. The solid residues were resolved with 1 mL of hexane fortified with 50 µL of previously made solution of methyl nonadecanoate (C19:0) (internal standard; ISTD) in hexane and kept at −80 °C until analysis.

#### 3.3.4. Tissue Fatty Acids Determination

The FFAs analysis in the obtained samples was performed at Focus GC coupled with a PolarisQ mass spectrometer (Thermo Fisher, Waltham, MA, USA). The carrier gas was helium (1 mL/min), and the injected volume of the sample was 1 µL. The temperature program of the oven was as follows: the initial temperature 50 °C (1 min), then 25 °C/min to 200 °C, and immediately 3 °C/min to 230 °C (held for 18 min). The injector was in split mode (50:1), while the temperatures of the injector, transfer line, and ion source were 250 °C, 260 °C, and 260 °C, respectively. The concentration of 13 fatty acids was investigated: myristic (tetradecanoic, C14:0), pentadecylic (pentadecanoic, C15:0), palmitic (hexadecanoic, C16:0), palmitoleic (cis-Δ9 hexadecenoic, C16:1), margaric (heptadecanoic, C17:0), stearic (octadiecanoic, C18:0), oleic/elaidic (cis- and trans-Δ9 octadecenoic, C18:1, cis + trans), linoleic/linolelaidic (cis- and trans-Δ9,12 octadecenoic, C18:2, cis + trans) arachidonic acid (cis-Δ5,8,11,14 eicosatetraenoic acid, C20:4), behenic (docosanoic, C22:0), erucic (cis,cis-Δ13,16 docosadienoic, C22:1), and cervonic (cis,cis,cis,cis,cis,cis-Δ4,7,10,13,16,19 docosahexaenoic, C22:6). Under the applied chromatographic conditions, cis- and trans- forms of C18:1 and C18:2 coelute at the same retention time, so their concentrations were calculated as the sum.

#### 3.3.5. Tissue Triglycerides Determination

The TGAs analysis was performed on Camag TLC Scanner 3. The 2 µL of lipid extract was applied to 20 × 20 cm HPTLC silica gel plates (Art. 105641, Merck) as 6-mm band by using an Automatic TLC sampler 4 (ATS4, CAMAG, Muttenz, Switzerland). The ascending chromatography was performed in the CAMAG twin-trough chamber using four mobile phases: chloroform:methanol:acetic acid (90:10:1, *v*/*v*/*v*) up to 25mm, n-hexane:diethyl-ether:acetone (60:40:5, *v*/*v*/*v*) up to 70 mm, n-hexane:diethyl-ether up to 85 mm, and 100% n-hexane up to 90mm. Before the analysis, the chambers were saturated for 30 min. After the last mobile phase development, the plates were dried in the dark. Derivatization was performed by spraying with a mixture of methanol and concentrated sulphuric acid (9/1, *v*/*v*). Derivatized plates were heated at 80 °C on TLC Plate Heater III (CAMAG) until chromatographic zones become visible, followed by scanning in CAMAG TLC scanner 3. TGAs concentration was determined based on the intensity of the standards and samples.

### 3.4. Determination of Oxidative Stress Biomarkers

The oxidative stress biomarkers were measured in tissue samples prepared as it was described above in the case of tissue AST, ALT, AP, and GGT activity measurement. The activity of SOD (EC 1.15.1.1) was assayed by the epinephrine method [85]. One unit of SOD activity was defined as the amount of protein causing 50% inhibition of the autoxidation of adrenaline to adrenochrome at 26 °C and was expressed as U/g wet mass. For the determination of MnSOD activity, the assay was performed after pre-incubation with 8 mmol/L KCN. CuZnSOD activity was calculated as the difference between the total SOD and MnSOD activities. CAT activity (EC 1.11.1.6) was evaluated by the rate of hydrogen peroxide decomposition [86] and expressed as U/g wet mass with the one unit defined as the µmol H_2_O_2_/min/g wet mass. The activity of GSH-Px (EC 1.11.1.9) was determined by following the oxidation of nicotinamide adenine dinucleotide phosphate (NADPH) with t-butyl hydroperoxide [87] and expressed as U/g wet mass with the one unit defined as the nmol NADPH/min/g wet mass. The concentration of thiobarbituric acid reactive substances (TBARS) was estimated according to the method of Rehncrona et al. [88]. The products of TBARS formed spontaneously, and red color was produced by the reaction of thiobarbituric acid (TBA) with lipid peroxidation products (malondialdehyde). The measurement was performed upon treating the samples with cold thiobarbituric acid reagent (40% trichloroacetic acid, 0.6% thiobarbituric acid) and subsequent heating at 100 °C and recorded at 532 nm. The results of LPO were expressed as nmol TBARS/g tissue.

### 3.5. Determination of Adrenaline and Noradrenaline Content

#### 3.5.1. Determination of Adrenaline and Noradrenaline Content in Serum

Serum catecholamines were determined commercially in the BelMedic laboratory, Belgrade, Serbia.

#### 3.5.2. Determination of Adrenaline and Noradrenaline Content in Adrenal Glands

For high performance liquid chromatography (HPLC) analysis, adrenal glands were dissected and immediately stored at −70 °C. Tissue samples were homogenized in DEPROT solution (1 mg:30 μL) containing 2% ethylene glycol tetra-acetic acid, 0.1 N HClO4, and 0.2% MgCl2, sonicated and centrifuged (30 min, 18000 rpm, 4 °C). Fifty µL of collected supernatants were injected with the autosampler of a Dionex UltiMate 3000 HPLC system (Thermo Scientific, Sunnyvale, CA, USA) equipped with an Acclaim Polar Advantage II (C18, 5 µm, 4.6 mm, 150 mm) HPLC column (Thermo Scientific, Waltham, Massachusetts, United States). The Chromeleon7 Chromatography Data System (Thermo Scientific, Sunnyvale, CA, USA) was used for instrument control and data acquisition. The mobile phase consisted of 98% ammonium formate buffer (Fisher Scientific, Cambridge, UK, pH 3.6) and 2% methanol (J.T. Baker, Griesheim, Germany), with the flow rate set at 500 μL/min. The electrochemical measurement was set at +850 mV potential and the separation temperature at 25 °C. Noradrenaline (DL-noradrenaline hydrochloride, Sigma-Aldrich) and adrenaline ((±)-adrenaline hydrochloride, Sigma-Aldrich) standard solutions were created from the stock standard solution (1 mg/mL of noradrenaline in methanol) in DEPROT, with concentration range 0.5–50 μg/mL.

### 3.6. Western Immunoblot Analysis

Liver, kidney, and heart tissue samples were homogenized using a handheld homogenizer (Ultra-Turrax, Sigma-Aldrich, USA) in ice-cold RIPA buffer (50 mM Tris-HCL pH = 7.44; 0.1% SDS; 150 mM NaCl; 10 mM EDTA; 10 mM EGTA; 1 % NP-40; 0.5% Triton-X) containing protease and phosphatase inhibitors (SigmaFAST protease inhibitor cocktail and Na-orthovanadate, respectively). Homogenates were centrifuged at 10,000× *g* (Sorvall SL-50T, Super T21, Thermo Fisher Scientific) for 20 min at 4 °C. Resulting supernatants were aliquoted and stored at −80 °C.

Protein samples of serum (1 µL) separated by 12% Tricine SDS-PAGE, and protein samples of whole liver, kidney, and heart homogenates (20 µg) were separated by 12% SDS-PAGE and transferred to PVDF membranes (BioRad, Hercules, CA, USA), blocked in TBST solution (0.2 % Tween 20, 50 mM Tris-HCl pH 7.6, 150 mM NaCl) containing 5% non-fat condensed milk, and incubated overnight, at 4 °C, with goat polyclonal anti-HMGB1 for serum analysis (1:1000, K-12; Santa Cruz Biotechnology, Dallas. TX, USA), rabbit polyclonal antibodies specific to HMGB1 (1:1000, ab18256; Abcam, Cambridge, UK), phospho-NF-ĸB p65 (1:750, ab 194926; Abcam), β-actin (1:1000, ab 8227; Abcam), Bax (1:1000, 2772; Cell Signalling), Bcl-2 (1:500, NB100-56098; Novus Biological), rabbit monoclonal (EPR22856-212) anti-Hp antibody (1:1000, ab256454; Abcam). PGC1-α was analyzed using rabbit polyclonal PGC1-α-antibody (1:1000, ab191838; Abcam), and using beta actin (1:30,000, ab49900; Abcam) for liver and kidney, and GAPDH-antibody (1:10,000, MAB374; Millipore Sigma, Burlington, MA, USA) for heart homogenates.

After washing in TBST solution, the blots were probed with horseradish peroxidase-conjugated secondary antibody purchased from Abcam, goat anti-rabbit IgG (1:2000, ab205718, ab 97051), or Santa Cruz Biotechnology (1:1000, sc 2378) for 2 h at 4 °C. Detection of immunoreactive bands was performed by an enhanced chemiluminescence detection system (Santa Cruz Biotechnology) using the iBright CL1500 Imaging System (Thermo Fisher Scientific, USA). For re-probing, membranes were incubated in 2% SDS, 100 mM β-mercaptoethanol, and 62.5 mM Tris-HCl pH 6.8 for 35 min at 50 °C, then rinsed three times, blocked, and probed again with another antibody. The quantification of immunoreactive bands was performed using the TotalLab (Phoretix, Newcastle Upon Tyne, UK) electrophoresis software (version 1.10).

### 3.7. Histology Analysis

Tissue samples from the heart, liver, and kidney were fixed in a 10% neutral buffered formalin for ~48, washed in a series of increasing ethanol solutions (70%, 96%, and 100%), and immersed in xylene. Afterwards, samples were embedded into paraffin wax and cut into 5 µm thick sections. Four sub-serial cross-sections of kidney were stained with periodic acid-Schiff (PAS) and Haematoxylin and Eosin stain (H&E stain), while samples of the heart and liver were stained with H&E stain only. A minimum of 10 fields of each organ section were analyzed by an Olympus BX43 microscope connected to a Leica ICC50W camera.

Heart tissue was evaluated using the Dallas criteria for the diagnosis of myocarditis [89]. The extent of inflammatory infiltrate was described either as global or focal with the patchy disease. The other pathohistological changes were also analyzed, such as hypertrophy, dilatation, vacuolar degeneration of myocytes (myocytolysis), and stasis.

Morphological changes in liver tissue were scored according to the Suzuki histological score from 0–4, based on the intensity of congestion, hepatocytes necrosis, cytoplasmatic vacuolization, and cellular morphology characteristics [90]. In addition, tissue organization, portal inflammation, Kupffer cell hyperplasia, hepatocellular steatosis, and the presence of binuclear cells were also investigated.

Renal damage was evaluated by global renal damage score of the following lesions: tubular epithelial (TF) cell flattening—1 point, brush border loss (BBL)—1 point, and 2 points for bleb formation (BF), tubular necrosis (TN) and tubular lumen obstruction (TO) [91].

### 3.8. Statistical Analysis

Differences in investigated parameters between the groups were calculated using one-way ANOVA. When significant differences were found, pairwise comparisons were performed using Holm-Sidak post hoc tests. Survival Analysis was performed by Kaplan-Meier survival analysis, using the LogRank test for the comparison of multiple curves. If the LogRank statistic showed a significant difference in survival curves, the Holm-Sidak test was used for the posterior comparison. Statistical package SIGMAPLOT was used for all the analyses and graphical presentations. The level of statistical significance was defined as *p* < 0.05. Where appropriate, analysis results were graphically presented as the percentage of the control (C) group.

## 4. Conclusions

The importance of energy production, expenditure, and maintenance in sepsis is a well-recognized phenomenon. However, there are not many data available on molecules altering energy homeostasis in an organism facing septic challenges. For the first time, we reported detrimental effects of meldonium, a therapeutic otherwise proven helpful in treating and preventing ischemia. The paradox of this finding lies in the observation that the same properties of meldonium, which allow it to act protectively in ischemia, actually lessen the survival chances under septic conditions. On that account, our results underline not only the importance of the uninterrupted energy production in sepsis, but also draw attention to the possible harmful effects of lipid-mobilization impairment present in certain clinical conditions or caused by certain therapeutics. This could lead to the much-needed revision of the existing guideline in the clinical treatment of sepsis while paving the way for discovering new therapeutic approaches.

## Figures and Tables

**Figure 1 ijms-22-09698-f001:**
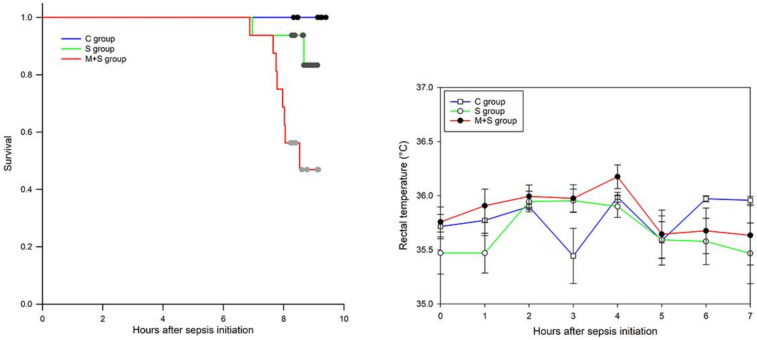
Survival rate analysis (left) and time-course curve of rectal temperature recordings (right) in rats of control (C), sepsis (S), and meldonium + sepsis (M + S) groups. The data on the rectal temperature recordings are given as time-points mean ± standard error.

**Figure 2 ijms-22-09698-f002:**
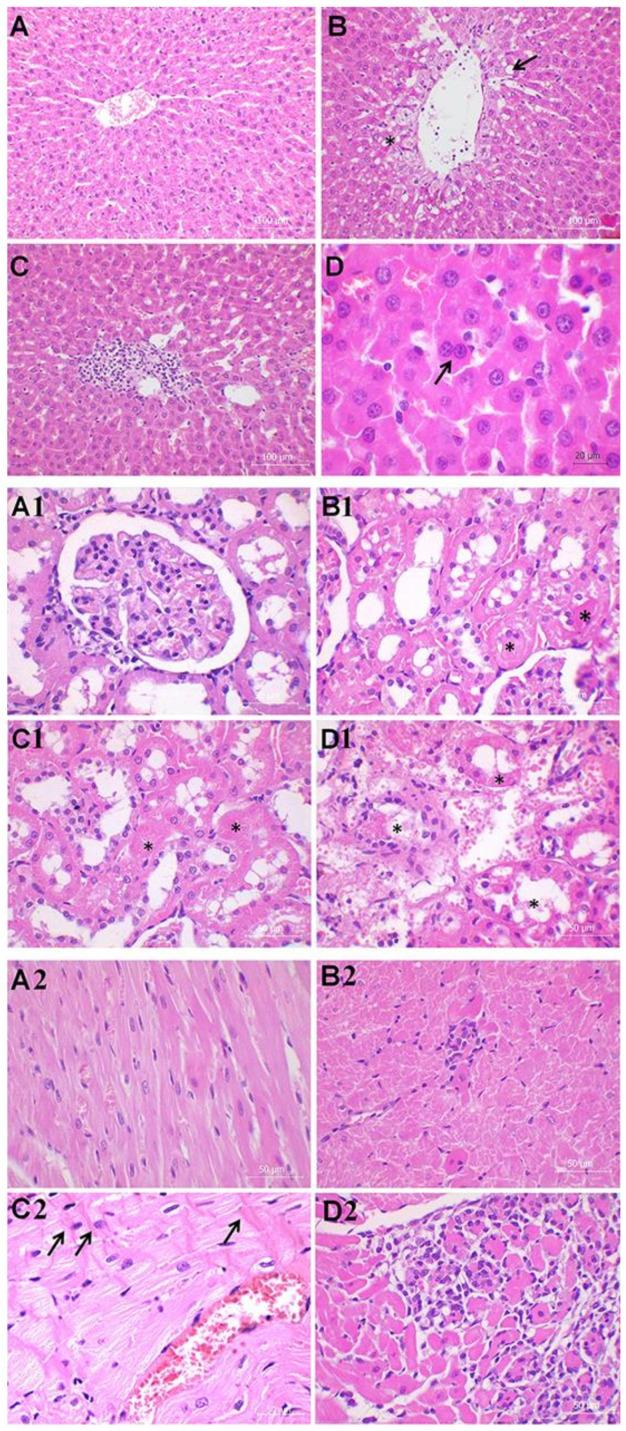
Liver histology analysis (40× and 60×). Group categories abbreviations: (**A**) control (**C**) rat group; (**B**,**C**) sepsis (S) rat group; (D) meldonium + sepsis (M + S) rat group. Results of liver analysis: (**A**) Normal histological structure of the hepatic tissue; (**B**) The black asterisk indicates the parenchymal vacuolization owing to the hydropic degeneration and cytoplasm swelling of the hepatocytes. The black arrow indicates steatosis; (**B**,**C**) Micrography presents necrosis of larger groups of hepatocytes (patchy necrosis) and sinusoidal congestion with the extravasation of erythrocytes; (**D**) The black arrow indicates binucleation of hepatocytes. Kidney histology analysis (40 ×). Group categories abbreviations: (**A1**) control (C) rat group; (**B1**) sepsis (S) rat group; (**C1,D1**) meldonium + sepsis (M + S) rat group. Results of kidney analysis: (A1) Normal histological structure of glomeruli and tubules; (**B1**) Micrography displays moderate kidney damage with the reduction or complete loss of the epithelial brush border. The black asterisk indicates rare bleb formation; (**C1,D1**) Severe tubular necrosis with dilatation of the tubular structure, tubular lumen obstruction, and partial loss of tissue architecture is presented. The black asterisk indicates tubular lumen obstruction. Heart histology analysis (40×). Group categories abbreviations: (**A2**) control (C) rat group; (**B2**) sepsis (S) rat group; (**C2**,**D2**) meldonium + sepsis (M + S) rat group. Results of heart analysis: (**A2**) Control rats showing normal histological structure; (**B2**) Micrography displays myocytolysis, interstitial mononuclear infiltration, and a loss of striation; (**C2**,**D2**) Micrography shows severe and diffuse interstitial mononuclear infiltration with greater loss of myocytes. Contraction band necrosis is characterized by the bright eosinophilic cross-bands within the cells (as indicated by the black arrows).

**Figure 3 ijms-22-09698-f003:**
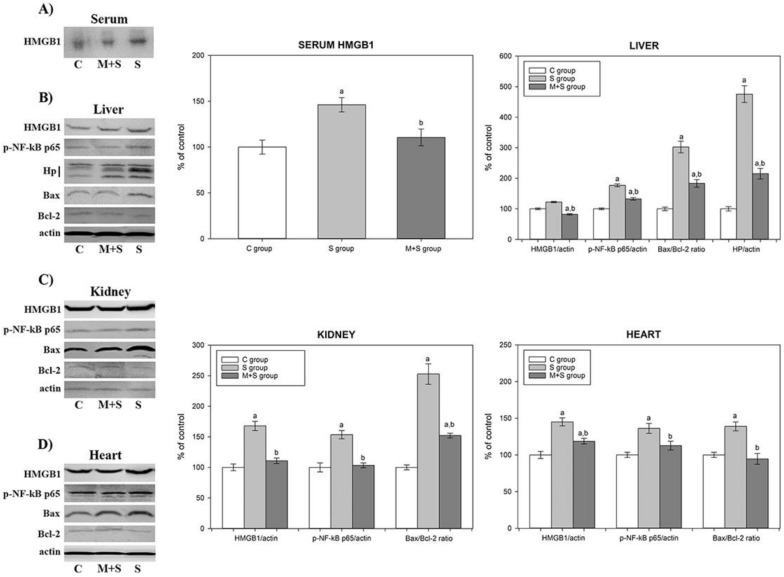
Representative immunoblots of protein expression levels: (**A**) serum HMGB1; (**B**) liver HMGB1, p-NF-κB p65, Hp, Bax, and Bcl-2; (**C**) kidney HMGB1, p-NF-κB p65, Bax, and Bcl-2; and (**D**) heart HMGB1, p-NF-κB p65, Bax, and Bcl-2 in rats of control (C), sepsis (S), and meldonium + sepsis (M + S) groups. β actin was used as a loading control. Protein expression levels were calculated and graphically presented as the percent of control (mean ± standard error). Note: tissue samples were loaded on the gels in the groups order C, M + S, and S (blots presented on the left side of the figures), whereas calculated protein expression levels were graphically presented in the groups’ order C, S, and M + S (graphs presented on the right side of the figure). Minimal significance level: *p* < 0.05. Significantly different: ^a^ in respect to C; ^b^ in respect to S.

**Figure 4 ijms-22-09698-f004:**
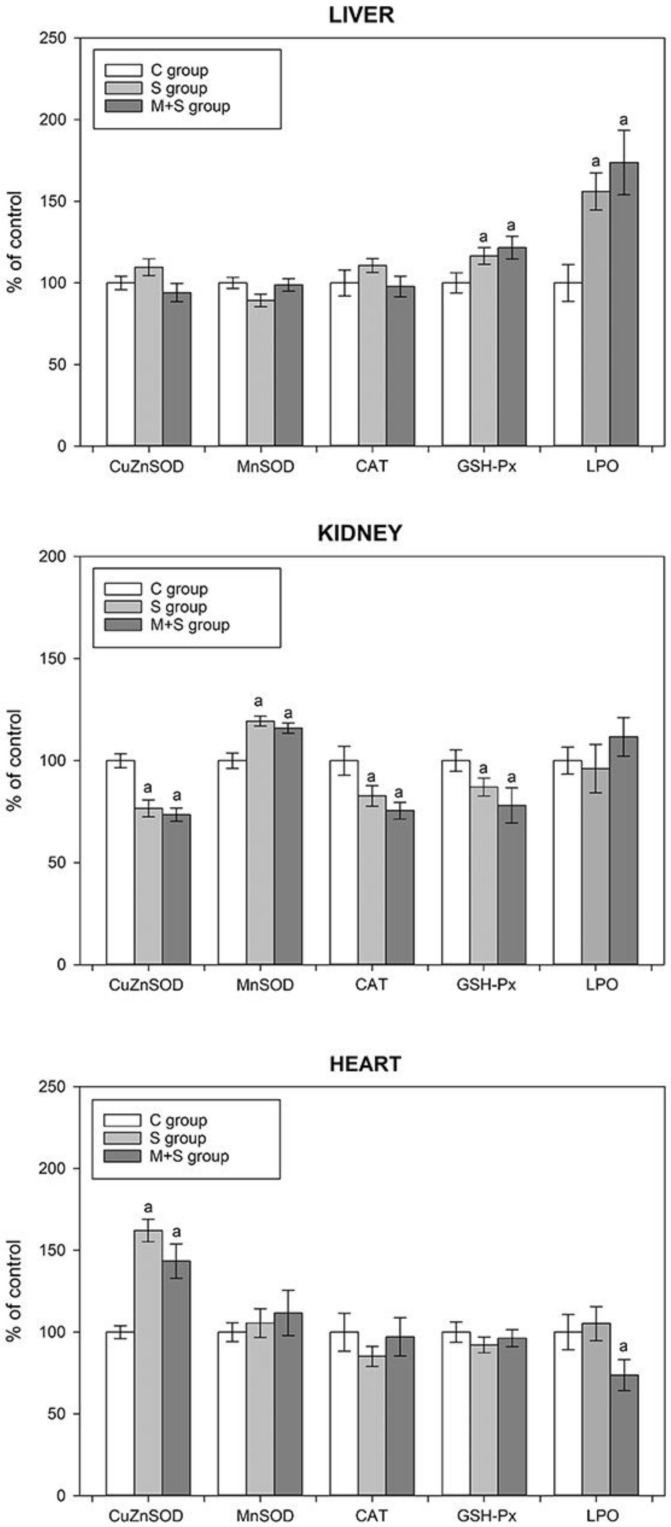
The tissue activities of liver copper-zinc superoxide dismutase (CuZnSOD), manganese superoxide dismutase (MnSOD), catalase (CAT), and glutathione peroxidase (GSH-Px), and tissue lipid peroxidation level (LPO) in the liver, kidney, and heart of rats of control (C), sepsis (S), and meldonium + sepsis (M + S) groups. The data are calculated and graphically presented as the percent of control (mean ± standard error). The number of animals per experimental group: n = 8. Minimal significance level: *p* < 0.05. Significantly different: ^a^ in respect to the C group.

**Figure 5 ijms-22-09698-f005:**
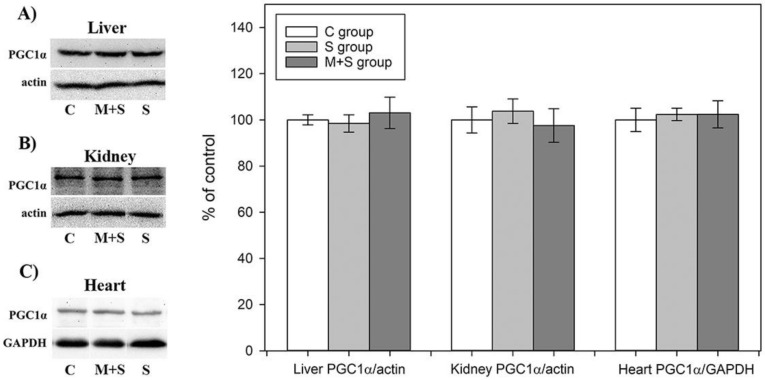
Representative immunoblots of protein expression levels of peroxisome proliferator-activated receptor gamma coactivator 1-alpha in the: (**A**) liver; (**B**) kidney; and (**C**) heart in rats of control (C), sepsis (S), and meldonium + Sepsis (M + S) groups. β actin was used as a loading control for the liver and kidney, while glyceraldehyde 3-phosphate dehydrogenase (GAPDH) was used as a loading control for the heart. Protein expression levels are calculated and graphically presented as the percent of control (mean ± standard error). Note: tissue samples were loaded on the gels in the groups order C, M + S, and S (blots presented on the left side of the figures), whereas calculated protein expression levels were graphically presented in the groups’ order C, S, and M + S (graphs presented on the right side of the figure). Minimal significance level: *p* < 0.05.

**Table 1 ijms-22-09698-t001:** The aspartate aminotransferase (AST), alanine aminotransferase (ALT), and alkaline phosphatase (ALP) activities (IU/L), and serum Troponin T concentration (ng/L) in rats of control (C), sepsis (S), and meldonium + sepsis (M + S) groups. The number of animals per experimental group: n = 8. The data are given as mean ± standard error of the mean. Minimal significance level: *p* < 0.05. Significantly different: ^a^ in respect to the C group; ^b^ in respect to the S group.

	C Group	S Group	M + S Group
AST	266.88 ± 19.34	345.17 ± 25.31 ^a^	243.50 ± 21.79 ^b^
ALT	102.13 ± 2.57	172.00 ± 12.00 ^a^	103.00 ± 5.59 ^b^
ALP	150.50 ± 5.67	227.83 ± 23.92 ^a^	244.50 ± 18.22 ^a^
Troponin T	53.00 ± 5.79	141.17 ± 3.07 ^a^	91.67 ± 5.19 ^a,b^

**Table 2 ijms-22-09698-t002:** Tissue histological scores in liver, kidney, and heart of rats of control (C), sepsis (S), and meldonium + sepsis (M + S) groups: Liver Suzuki histological score (0–4) based on the sinusoidal congestion, vacuolization of hepatocyte cytoplasm, and parenchymal necrosis (0—no changes, 1—minimal changes, 2—mild changes, 3—moderate changes, 4—severe changes); Renal histological score (0–4) based on the tubular epithelial cell flattening (TF), brush border loss (BBL), bleb formation (BF), tubular necrosis (TN), and tubular lumen obstruction (TO) (0—no changes, 1—minimal changes, 2—mild changes, 3—moderate changes, 4 or greater—severe changes); Heart histological score (Dallas criteria) based on the extent of inflammatory infiltration (II) (0—no changes, 1—mild changes, 2—moderate changes, 3—severe changes), and its distribution (1—focal, 2—confluent, 3—diffuse).

**Liver**	**C Group**	**S Group**	**M + S Group**
Congestion	1	1.333	1.125
Vacuolization	0.375	1.333	0.625
Necrosis	0.250	1.417	1.125
**Kidney**	**C group**	**S group**	**M + S group**
TF	2.875	7.333	8.250
BBL	2.250	6.583	9.500
BF	2.125	6.500	15.500
TN	1.375	7.000	14.750
TO	1.375	6.000	14.500
**Heart**	**C group**	**S group**	**M + S group**
II	0.250	0.583	1.750
Distribution	0	0.333	1.250

**Table 3 ijms-22-09698-t003:** Adrenal glands and serum adrenaline and noradrenaline concentrations and liver, kidney, and heart lipidomics in rats of control (C), sepsis (S), and meldonium + sepsis (M + S) groups. All units are given as mg/kg tissue. FFAs abbreviations: myristic—C14:0; pentadecylic—C15:0; palmitic—C16:0; palmitoleic—C16:1; margaric—C17:0; stearic—C18:0; oleic/elaidic—C18:1 cis + trans; linoleic/linolelaidic—C18:2, cis + trans; arachidonic—C20:4; behenic—C22:0; erucic—C22:1; cervonic—C22:6. The number of animals per experimental group: n = 8. The data are given as mean ± standard error of the mean. Minimal significance level: *p* < 0.05. Significantly different: ^a^ in respect to the C group; ^b^ in respect to the S group.

**Adrenal Glands**	**C Group**	**S Group**	**M + S Group**
Adrenaline	100 ± 4.945	60.586 ± 2.946 ^a^	29.058 ± 0.964 ^a,b^
Noradrenaline	100 ± 3.221	98.356 ± 2.389	83.698 ± 3.232 ^a,b^
**Serum**	**C group**	**S group**	**M + S group**
Adrenaline	1651.983 ± 66.585	1238.067 ± 53.207 ^a^	1354.9 ± 17.081 ^a^
Noradrenaline	1297.667 ± 21.243	1475.2 ± 24.198 ^a^	957.233 ± 46.875 ^a,b^
**Liver**	**C group**	**S group**	**M + S group**
L carnitine	6.600 ± 0.024	17.256 ± 0.922 ^a^	4.631 ± 0.052 ^a,b^
Glycerol	0.071 ± 0.001	1.231 ± 0.019 ^a^	0.879 ± 0.014 ^a,b^
Triglycerides	1.372 ± 0.085	5.750 ± 0.078 ^a^	2.052 ± 0.070 ^a,b^
Glucose	2.353 ± 0.021	3.459 ± 0.045 ^a^	2.588 ± 0.024 ^b^
Lactate	1.936 ± 0.011	4.194 ± 0.146 ^a^	2.469 ± 0.017 ^a,b^
Fructose	0.024 ± 0.001	0.888 ± 0.006 ^a^	0.081 ± 0.001 ^a,b^
Sucrose	0.021 ± 0.001	0.108 ± 0.005 ^a^	0.061 ± 0.001 ^a,b^
C14:0	8.333 ± 0.078	13.358 ± 0.400 ^a^	6.393 ± 0.062 ^a,b^
C15:0	6.363 ± 0.041	11.768 ± 0.326 ^a^	7.248 ± 0.029 ^b^
C16:0	5.145 ± 0.024	10.205 ± 0.304 ^a^	4.681 ± 0.034 ^b^
C16:1	0.461 ± 0.011	0.650 ± 0.009 ^a^	0.307 ± 0.012 ^a,b^
C17:0	0.346 ± 0.036	0.723 ± 0.003 ^a^	0.309 ± 0.012 ^b^
C18:0	5.121 ± 0.008	12.429 ± 0.127 ^a^	4.779 ± 0.032 ^b^
C18:1, c + t	2.175 ± 0.025	3.030 ± 0.024 ^a^	1.519 ± 0.030 ^a,b^
C18:2, c + t	5.013 ± 0.043	9.461 ± 0.145 ^a^	4.265 ± 0.027 ^b^
C20:4	5.484 ± 0.112	12.267 ± 0.122 ^a^	5.633 ± 0.093 ^b^
C22:0	1.505 ± 0.037	1.340 ± 0.056 ^a^	0.832 ± 0.010 ^a,b^
C22:1	0.230 ± 0.008	0.395 ± 0.027 ^a^	0.313 ± 0.007 ^a,b^
C22:6	0.825 ± 0.005	1.565 ± 0.104 ^a^	0.845 ± 0.012 ^b^
The FFAs sum	40.951 ± 0.263	77.189 ± 0.917 ^a^	37.124 ± 0.191 ^b^
**Kidney**	**C group**	**S group**	**M + S group**
L carnitine	1.757 ± 0.017	8.512 ± 0.664 ^a^	1.551 ± 0.179 ^b^
Glycerol	0.372 ± 0.007	0.671 ± 0.018 ^a^	0.317 ± 0.015 ^b^
Triglycerides	1.624 ± 0.049	1.107 ± 0.013 ^a^	1.278 ± 0.022 ^a^
Glucose	1.275 ± 0.007	3.309 ± 0.098 ^a^	0.700 ± 0.012 ^a,b^
Lactate	1.932 ± 0.128	3.473 ± 0.179 ^a^	2.719 ± 0.158 ^a,b^
Fructose	0.117 ± 0.055	0.573 ± 0.113 ^a^	0.104 ± 0.056 ^b^
Sucrose	0.051 ± 0.001	0.103 ± 0.010 ^a^	0.060 ± 0.001 ^b^
C14:0	6.054 ± 0.123	37.325 ± 1.935 ^a^	16.685 ± 0.260 ^a,b^
C15:0	82.960 ± 2.172	3.288 ± 0.861 ^a^	0.127 ± 0.006 ^a,b^
C16:0	39.500 ± 1.617	489.563 ± 25.487 ^a^	143.866 ± 3.856 ^a,b^
C16:1	56.845 ± 1.250	11.740 ± 0.928	0.095 ± 0.002 ^a,b^
C17:0	59.144 ± 1.534	24.153 ± 3.155 ^a^	0.098 ± 0.002 ^a,b^
C18:0	44.908 ± 1.154	172.758 ± 17.153 ^a^	50.237 ± 1.445 ^b^
C18:1 cis + trans	18.642 ± 0.852	103.390 ± 14.797 ^a^	0.260 ± 0.011 ^a,b^
C18:2 cis + trans	24.550 ± 0.601	82.404 ± 14.128 ^a^	20.061 ± 0.664 ^b^
C20:4	46.280 ± 1.108	0.499 ± 0.0267 ^a^	0.523 ± 0.025 ^a^
C22:0	20.489 ± 0.699	0.825 ± 0.105 ^a^	0.836 ± 0.018 ^a^
C22:1	26.175 ± 1.109	7.219 ± 0.315 ^a^	11.429 ± 0.483 ^a,b^
C22:6	17.587 ± 1.102	5.534 ± 0.295 ^a^	13.175 ± 0.481 ^b^
The FFAs sum	443.134 ± 11.702	938.697 ± 66.007 ^a^	257.395 ± 4.710 ^a,b^
**Heart**	**C group**	**S group**	**M + S group**
L carnitine	0.990 ± 0.012	3.976 ± 0.042 ^a^	0.760 ± 0.008 ^b^
Glycerol	0.074 ± 0.018	0.307 ± 0.014 ^a^	0.088 ± 0.011 ^b^
Triglycerides	0.856 ± 0.033	1.959 ± 0.024 ^a^	0.952 ± 0.017 ^b^
Glucose	0.793 ± 0.006	0.554 ± 0.005 ^a^	0.129 ± 0.002 ^a,b^
Lactate	0.309 ± 0.001	1.316 ± 0.014 ^a^	0.305 ± 0.001 ^b^
Fructose	0.012 ± 0.001	0.012 ± 0.001	0.0085 ± 0.0001 ^a,b^
Sucrose	0.0012 ± 0.0001	0.0065 ± 0.0002 ^a^	0.0041 ± 0.0001 ^a,b^
C14:0	7.194 ± 0.074	25.342 ± 0.874 ^a^	11.539 ± 0.130 ^a,b^
C15:0	44.661 ± 1.097	7.528 ± 0.409 ^a^	3.688 ± 0.0142 ^a,b^
C16:0	22.322 ± 0.809	249.884 ± 12.715 ^a^	74.273 ± 1.922 ^a,b^
C16:1	28.653 ± 0.626	6.195 ± 0.462 ^a^	0.201 ± 0.005 ^a,b^
C17:0	29.745 ± 0.774	12.438 ± 1.578 ^a^	0.204 ± 0.006 ^a,b^
C18:0	25.015 ± 0.580	92.593 ± 8.586 ^a^	27.508 ± 0.718 ^b^
C18:1 cis + trans	10.408 ± 0.421	53.210 ± 7.404 ^a^	0.890 ± 0.014 ^a,b^
C18:2 cis + trans	14.782 ± 0.290	45.933 ± 7.043 ^a^	12.163 ± 0.324 ^b^
C20:4	25.882 ± 0.558	6.383 ± 0.056 ^a^	3.078 ± 0.041 ^a,b^
C22:0	10.997 ± 0.357	1.083 ± 0.044 ^a^	0.835 ± 0.008 ^a^
C22:1	13.202 ± 0.557	3.807 ± 0.150 ^a^	5.871 ± 0.239 ^a,b^
C22:6	9.205 ± 0.550	3.549 ± 0.144 ^a^	7.010 ± 0.241 ^a,b^
The FFAs sum	242.067 ± 5.894	507.943 ± 32.945 ^a^	147.260 ± 2.356 ^a,b^

**Table 4 ijms-22-09698-t004:** The study designs. C—control (sham) group of animals (free access to tap water for four weeks, and then received a saline injection); S—septic group of animals (free access to tap water for four weeks, and then received a faecal intraperitoneal injection); and M + S—meldonium-septic group of animals (free access to tap water with meldonium for four weeks, and then received a faecal intraperitoneal injection). The number of animals per group = 8.

Groups	C Group	S Group	M + S Group
Application of meldonium in water	none	none	+
Intraperitoneal application of saline	+	none	none
Intraperitoneal application of faeces	none	+	+

## Data Availability

The data presented in this study are available on request from the corresponding author.

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
