# Peer review of "The Effects of a Meldonium Pre-Treatment on the Course of the Faecal-Induced Sepsis in Rats"

_ijms, 2021, doi:10.3390/ijms22189698_

Round 1
Reviewer 1 Report
In the present paper Durasevic and colleagues tested the effect of meldonium in a rat model of fecal-induced sepsis. Given that energy availability is an important prognostic assessment of sepsis and that meldonium exerts a metabolic effect, they hypothesize a protective role of this molecule. Surprisingly, authors describe an increase in mortality rate of sepsis-induced rat pre-treated with meldonium. They suggested the overall decrease in lipid metabolism as the main adverse effect accounting for the harmful effect of meldonium.
Although the topic is of interest, some issues need to be addressed before considering the paper suitable for publication.
MAIO ISSUES
- Introduction paragraph need to be revised. Authors should better define the specific hypotheses being tested. Given the controversial results obtained, their hypotheses need to be better argued. A hint about metabolic modulatory activity of meldonium, defined as performance-enhancing drug, could provide a more comprehensive overview on the topic (see PMID: 30362565).
- It would be more informative if the authors could provide a scheme representing the experimental plan used in their study. The final part of introduction could be used in the incipit of results sub-paragraphs (see point 3 and 4).
- Manuscript section need to follow the scheme provided in authors’ guideline. Accordingly, authors must rearrange the manuscript and slip the paragraph “Result and Discussion” in two separate ones (i.e., Results paragraph and discussion paragraph). Please see https://www.mdpi.com/files/word-templates/ijms-template.dot
- Results should be arranged in sub-sections in order to better define the results obtained. Analogously the discussion paragraph needs to follow the same scheme.
- In figure 3B the expression level of HMG1 in liver homogenate of the S group seem to be equal to control. Please check and correct in the text.
- Given the authors proposed the metabolic alterations as the explanation of the increased mortality rate of M+G group, it would be more informative if the they could look at the mitochondrial status of the groups analyzed. For instance, authors could assay the expression levels of Peroxisome proliferator-activated receptor gamma coactivator-1 alpha (PGC-1α), or of other markers (see 18430751; 20933024; 30362565)
MINOR ISSUES
- Authors should provide indication about the statistics used for survival analysis.
- Figure 3 needs to be rearranged. Please pair the western blot images with the relative histograms
Author Response
Dear Sir/Madam,
Thank you for the useful comments. We did our best to correct/improve the manuscript accordingly. Our answers are provided below, and the changes made in the original manuscript are highlighted in yellow.
Reviewer 1:
In the present paper Djurasevic and colleagues tested the effect of meldonium in a rat model of faecal-induced sepsis. Given that energy availability is an important prognostic assessment of sepsis and that meldonium exerts a metabolic effect, they hypothesize a protective role of this molecule. Surprisingly, authors describe an increase in mortality rate of sepsis-induced rat pre-treated with meldonium. They suggested the overall decrease in lipid metabolism as the main adverse effect accounting for the harmful effect of meldonium. Although the topic is of interest, some issues need to be addressed before considering the paper suitable for publication.
MAIN ISSUES:
Point 1. Introduction paragraph need to be revised. Authors should better define the specific hypotheses being tested. Given the controversial results obtained, their hypotheses need to be better argued. A hint about metabolic modulatory activity of meldonium, defined as performance-enhancing drug, could provide a more comprehensive overview on the topic (see PMID: 30362565).
Response 1. According to the proposal, we reworked the previous manuscript and added new elements to the Introduction, which we hope will better define the specific hypothesis of the experiment (please see Lines 47-50 and 55-77).
Point 2. It would be more informative if the authors could provide a scheme representing the experimental plan used in their study. The final part of introduction could be used in the incipit of results sub-paragraphs (see point 3 and 4).
Response 2. As suggested, we have added a scheme representing key points of the experimental design used in our study (please see Table 4, Material and methods section, Lines 549-554). Additionally, we have arranged the Results and Discussion section into the corresponding sub-paragraphs.
Point 3. Manuscript section need to follow the scheme provided in authors’ guideline. Accordingly, authors must rearrange the manuscript and slip the paragraph “Result and Discussion” in two separate ones (i.e., Results paragraph and discussion paragraph). Please see https://www.mdpi.com/files/word-templates/ijms-template.dot.
Response 3. We followed the journal guideline provided in the “Research Manuscript Sections” at https://www.mdpi.com/journal/ijms/instructions which allows authors to present Results and Discussion as one section, quote Authors should discuss the results and how they can be interpreted in perspective of previous studies and of the working hypotheses. The findings and their implications should be discussed in the broadest context possible and limitations of the work highlighted. Future research directions may also be mentioned. This section may be combined with Results.
Point 4. Results should be arranged in sub-sections in order to better define the results obtained. Analogously the discussion paragraph needs to follow the same scheme.
Response 4. As stated in Response 2, we have arranged the Results and Discussion section into the corresponding sub-paragraphs.
Point 5. In figure 3B the expression level of HMG1 in liver homogenate of the S group seem to be equal to control. Please check and correct in the text.
Response 5. We apologize for the error. The error occurred on the graph of Figure 3B which has now been modified to correspond to the accompanying text.
Point 6. Given the authors proposed the metabolic alterations as the explanation of the increased mortality rate of M+G group, it would be more informative if the they could look at the mitochondrial status of the groups analyzed. For instance, authors could assay the expression levels of Peroxisome proliferator-activated receptor gamma coactivator-1 alpha (PGC-1α), or of other markers (see 18430751; 20933024; 30362565).
Response 6. As suggested, we blotted PGC-1α for the protein level analysis in all three tissues. The results are added as Figure 5 in the Results and Discussion section, along with the accompanying explanation (please see Lines 482-524).
MINOR ISSUES:
Point 1. Authors should provide indication about the statistics used for survival analysis.
Response 1. According to the proposal, we indicated the statistical test used for the survival analysis in the section Materials and Methods (please see Lines 779-782), as well as the results of the analysis in the section Results and Discussion (please see Lines 108-109).
Point 2. Figure 3 needs to be rearranged. Please pair the western blot images with the relative histograms
Response 2. The experimental design of our study imposes that a logical way of presenting the results would be arranging them in groups’ order: C (Control), S (sepsis induction) and M+S (meldonium pre-treatment + sepsis induction). Unfortunately, while performing the western blot, samples were loaded on the gels in the groups’ order C, M+S and S. Correcting this issue in the image processing programs would be considered as unallowed image manipulation. Therefore, to clarify on this matter and meet the reviewer’s suggestion, we have added notes in Figures 3 and 5, quote Note: tissue samples were loaded on the gels in the groups’ order C, M+S and S (blots presented on the left side of the figures), whereas calculated protein expression levels were graphically presented in the groups’ order C, S and M+S (graphs presented on the right side of the figure).
Reviewer 2 Report
1.Adrenal Gland functional assessments
- The manuscript states that adrenal gland function was evaluated based on adrenaline and noradrenaline. This would only assess medullary function of the adrenal gland but would not assess the similarly important and hypoxemia sensitive Cortex of the adrenal gland. Important hormones such as Cortisol and aldosterone are synthesized. Cortisol and aldosterone metabolism are important clinically in shock states as these are important for other aspects of cell stability and vascular tone.
- Assessments of the lipid metabolism should not only look at the long fatty acids but also at several components important for the overall hormonal metabolism such as cortisol and aldosterone. Looking at these may give the explanation why in the meldonium pretreated group especially the heart and the kidney were affected.
2. Labeling of the tables and graphs
- it is not clear when the assessments shown in the graphs and tables took place. It is assumed that these took place when the animals had been sacrificed, but this should be clearly stated in the labels. These tables and graphs should be supplemented by baseline measurements which can be provided in supplemental table. For experimental set-up the readers need to understand that the animals at the start of the different phases of the experiment where the same or already different in markers and histology.
-Pre-Pretreatment versus end of Meldonium pretreatment phase
-End of Meldonium pretreatment phase - start of shock initiation
-End of Meldonium pretreatment phase - End of treatment
-End of treatment versus death/ animal sacrifice
Information of these different phases will help understand the difference between Meldonium treatment and Meldnium/ Shock versus Sham treatment.
Author Response
Dear Sir/Madam,
Thank you for the useful comments. We did our best to correct/improve the manuscript accordingly. Our answers are provided below, and the changes made in the original manuscript are highlighted in yellow.
Reviewer 2:
Point 1. Adrenal Gland functional assessments
The manuscript states that adrenal gland function was evaluated based on adrenaline and noradrenaline. This would only assess medullary function of the adrenal gland but would not assess the similarly important and hypoxemia sensitive Cortex of the adrenal gland. Important hormones such as Cortisol and aldosterone are synthesized. Cortisol and aldosterone metabolism are important clinically in shock states as these are important for other aspects of cell stability and vascular tone. Assessments of the lipid metabolism should not only look at the long fatty acids but also at several components important for the overall hormonal metabolism such as cortisol and aldosterone. Looking at these may give the explanation why in the meldonium pre-treated group especially the heart and the kidney were affected.
Response 1. Unfortunately, due to the sample material lack, we were not able to include serum and/or adrenal aldosterone and cortisol analysis (more precisely: corticosterone, given the used experimental animal). Bearing in mind that rat adrenal glands are very small, we did not have enough tissue material as it was previously consumed for the catecholamine analysis. As opposed to the catecholamine analysis, a corticosteroid analysis cannot be performed on the HPLC. Namely, commercial kits must be used which require a larger amount of sample. In addition, given the importance of sympathoadrenal stimulation for energy production in general, we strongly believe that our decision to measure catecholamines instead of corticosteroids is a scientifically correct choice.
Point 2. Labelling of the tables and graphs
It is not clear when the assessments shown in the graphs and tables took place. It is assumed that these took place when the animals had been sacrificed, but this should be clearly stated in the labels. These tables and graphs should be supplemented by baseline measurements which can be provided in supplemental table. For experimental set-up the readers need to understand that the animals at the start of the different phases of the experiment where the same or already different in markers and histology.
-Pre-Pretreatment versus end of Meldonium pretreatment phase
-End of Meldonium pretreatment phase - start of shock initiation
-End of Meldonium pretreatment phase - End of treatment
-End of treatment versus death/ animal sacrifice
Information of these different phases will help understand the difference between Meldonium treatment and Meldonium/ Shock versus Sham treatment
Response 2. The proposed experimental design would require the usage of a far greater number of animals. This approach, even though scientifically justified, is not common for two main reasons. Firstly, a standard design of the experiment in which the possible protective effect of the substance of interest is studied involves a sham control group of animals, then a group of animals subjected to a model of given disorder or disease, and finally, a group of animals subjected to the same model of disorder or disease treated with the substance of interest. Secondly, usage of large number of animals in an experiment not followed by a justification of such approach entails the possibility of non-compliance by the Ethics Committee roles. Finally, as we have further clarified in the paper, we have already included sham+meldonium groups in our previous studies and observed no negative effect of meldonium (please see added explanation on the sham effects of meldonium in the Results and Discussion section, Lines 514-524).
Reviewer 3 Report
It is interesting study to investigate the effects of meldonium treatment on sepsis model.
The results of this study is complex, and were not explained in this study in terms of pathophysiological aspect.
The results of ALT, TnT, NF-kB, etc. are favoring the use of meldonium, but mortality rate and histological injury of organs such as kidney and heart revealed the opposite results.
It needs more data to explain this discrepancy.
They should at least infer the MOA of harmful effects of meldonium in sepsis.
For example, meldonium is FAO inhibitor, and in sepsis, oxidation of FFAs provides a crucial source of energy. With this background, the pretreatment of meldonium could be harmful in sepsis.
They used 3 groups, and what if meldonium has adverse effects on kidney in normal rats? To know that, meldonium pre-treatment only group is needed.
How about measuring serum creatinine level, NGAL, etc. as a marker of AKI?
Acute kidney injury was more severe in M+S group, and this could lead to metabolic acidosis, causing death. How about checking arterial blood gas analysis?
What is your explanation about discrepancy btw Tn and pathological scoring of heart?
Author Response
Dear Sir/Madam,
Thank you for the useful comments. We did our best to correct/improve the manuscript accordingly. Our answers are provided below, and the changes made in the original manuscript are highlighted in yellow.
Reviewer 3:
It is interesting study to investigate the effects of meldonium treatment on sepsis model. The results of this study are complex, and were not explained in this study in terms of pathophysiological aspect. The results of ALT, TnT, NF-kB, etc. are favouring the use of meldonium, but mortality rate and histological injury of organs such as kidney and heart revealed the opposite results. It needs more data to explain this discrepancy.
MAIN ISSUES:
Point 1. They should at least infer the MOA of harmful effects of meldonium in sepsis. For example, meldonium is FAO inhibitor, and in sepsis, oxidation of FFAs provides a crucial source of energy. With this background, the pre-treatment of meldonium could be harmful in sepsis.
Response 1. Given the so far published effects of meldonium, including its possible upregulating effect on the peroxisome proliferator-activated receptor gamma coactivator-1 alpha (PGC-1α) accompanied by the increased mitochondrial biogenesis (please see new results included given another reviewer’s suggestion in Figure 5 in the Results and Discussion section, Lines 472-513), our working hypothesis was that meldonium would act protectively, just as stated in the Introduction. In addition, there is a matter of increased lipolysis in sepsis, with the released free fatty acids (FFA) being taken up by the cells and subject to beta-oxidation. Meldonium blocks the carnitine palmitoyltransferase-1 (CPT1), leading to the accumulation of FFA and Acyl-CoA in the cytosol; which is a signal for the increased expression of PPARα and PGC-1α, further leading to the increased beta-oxidation of FFA and Acyl-CoA in peroxisomes (Berlato and Bairros, 2020, https://doi.org/10.1177/2397847320915143). All of the above considered, due to the redirection of FFA from mitochondria to peroxisomes, the harmful action of meldonium in sepsis cannot be automatically expected. For instance, based on our results, meldonium lowers lactate levels, which is an important prognostic marker of the improved sepsis course.
Point 2. They used 3 groups, and what if meldonium has adverse effects on kidney in normal rats? To know that, meldonium pre-treatment only group is needed.
Response 2. We did not provide a Sham + Meldonium group in this experiment, since the experimental design including Control, Sepsis and Meldonium + Sepsis fully corresponds to the usual concept of such experiments. In addition, the recent literature data on the pharmacological-toxicological profile of meldonium did not reveal any significant toxicity (Berlato and Bairros, 2020; https://doi.org/10.1177/2397847320915143). However, since we did include such groups in our previous studies , we added a corresponding explanation in the Results and Discussion section (please see Lines 514-524).
Point 3. How about measuring serum creatinine level, NGAL, etc. as a marker of AKI?
Response 3. Unfortunately, we witnessed that in septic rats, either meldonium-pretreated or not, it was highly difficult to withdraw enough blood probably due to the excessive fluid loss through other routes (e.g., via the peritoneum). Therefore, we weren’t able to perform additional analysis such as analysis of creatinine and/or NGAL level. However, as presented in the section regarding the histological analysis, detrimental changes were not limited to the kidneys only. Therefore, we assume that the presented results are sufficient to indicate that, given the worsened energy production, a specific combination of sepsis-induction and meldonium-pretreatment results in the increased mortality of the meldonium-pretreated animals.
Point 4. Acute kidney injury was more severe in M+S group, and this could lead to metabolic acidosis, causing death. How about checking arterial blood gas analysis?
Response 4. Unfortunately, due to the lack of the required equipment and expertise, we were not able to perform such analysis.
Point 5. What is your explanation about discrepancy btw Troponin T and pathological scoring of heart?
Response 5. We added a possible explanation on this matter in the Results and Discussion section (please see Lines 294-301).
Round 2
Reviewer 1 Report
The authors properly addressed the issues move
Reviewer 2 Report
Thank you for incorporating requested reviewer changes.
Reviewer 3 Report
Even though it is not sufficient answer for reviewer's question, authors tried to do their best with limited circumstances where they could not perform further animal study.